# Connecting higher education to workplace activities and earnings

**Hung Chau[1], Sarah H. Bana[2,3], Baptiste Bouvier[4], Morgan R. Frank** [1,3,5]*

**1** Department of Informatics and Networked Systems, University of Pittsburgh, Pittsburgh, PA, United States of America, **2** Argyros School of Business and Economics, Chapman University, Orange, CA, United States of America, **3** Digital Economy Lab, Institute for Human-Centered Artificial Intelligence, Stanford University, Stanford, CA, United States of America, **4** Computer Science Department, Massachusetts Institute of Technology, Cambridge, MA, United States of America, **5** Media Laboratory, Massachusetts Institute of Technology, Cambridge, MA, United States of America

* mrfrank@pitt.edu

## Abstract

Higher education is a source of skill acquisition for many middle- and high-skilled jobs. But what specific skills do universities impart on students to prepare them for desirable careers? In this study, we analyze a large novel corpora of over one million syllabi from over eight hundred bachelors' granting US educational institutions to connect material taught in higher education to the detailed work activities in the US economy as reported by the US Department of Labor. First, we show how differences in taught skills both within and between college majors correspond to earnings differences of recent graduates. Further, we use the co-occurrence of taught skills across all of academia to predict the skills that will be taught in a major moving forward. Our unified information system connecting workplace skills to the skills taught during higher education can improve the workforce development of high-skilled workers, inform educational programs of future trends, and enable employers to quantify the skills of potential workers.

## 1 Introduction

Education plays a critical role in economic growth and social progress. College degrees are generally associated with higher potential lifetime earnings, larger professional networks, and more adaptable careers [1, 2]. Higher education is a major part of US workforce development but information on the skills and expertise taught during higher education remain absent— even as recent research highlights the critical role of skills in shaping labor trends [3–5]. However, most empirical work relies on coarse labor distinctions, such as college major and institutional information (e.g., school brands), to explain these occupational trends [6–9]. While useful, these coarse educational and labor categories may hide further insights into the skills of "high-skilled" workers that contribute to positive career outcomes [10].

Many workers acquire skills through higher education that shape their careers. Studies have shown that social-cognitive skills and sensory-physical skills are correlated to high- and low-wage occupations, respectively, and that skill polarization divides workers with and without

**Funding:** This research is supported in part by the University of Pittsburgh Pitt Momentum Fund and the Center for Research Computing. This work has been supported (in part) by # 2109-33808 from the Russell Sage Foundation. Any opinions expressed are those of the principal investigator(s) alone and should not be construed as representing the opinions of the Foundation. The funders had no role in study design, data collection and analysis, decision to publish, or preparation of the manuscript.

**Competing interests:** The authors have declared that no competing interests exist.

higher education [11]. Discrepancies between skills demanded, taught, and researched have been identified by applying textual matching techniques to job advertisements, course syllabi, and research publications in Computer Science [12]. These analyses of skills reveal gaps between the workforce and educational/training systems. Understanding the sources of these gaps, across all fields of study, may improve curriculum design, inform educational policy, and improve student outcomes when they enter the workforce.

In this work, we analyze the recently-available Open Syllabus Project (OSP) dataset, which contains over 1.4 million course syllabi from more than 3,000 US colleges and universities from 2008 to 2017. While relatively new, this data source has proven useful for modeling higher education. For example, one study quantified the skill (mis-)alignment between academic research, industry, and educational offerings in data science and data engineering [12]. They used Burning Glass (BG) skill taxonomy and applied matching techniques to extract skills appearing in job titles and descriptions, course syllabi, and publication titles and abstracts. Another study proposed a new measure for the "education-innovation gap" using the textual similarity between course syllabi and academic journals to model the dissemination of frontier knowledge into college classrooms while relating these dynamics to students' graduation rates and incomes [13].

Our work is the first attempt to connect workplace activities to higher education through course syllabi; here, we use the granular workplace activities designed and produced by the U.S. Department of Labor (i.e., O*NET Detailed Work Activity (DWA) taxonomy described in Section Materials) to explain the underlying knowledge structures across college majors (*i.e.*, fields of study (FOS)) and among US universities. We use word embeddings to represent textual documents [14, 15], and explore different distance metrics to measure the similarity of two embedded skill vectors. Consequently, we are able to apply agglomerative hierarchical clustering techniques to the DWA-based vector representations of FOS and universities to discover their clusters. Hierarchical clustering [16] produces a nested sequence of cluster, and the hierarchy of clusters enables us to explore clusters at any level of detail without the need of identifying a specific number of topics as would be the case with K-means clustering techniques. Motivated by the principle of relatedness [17], we model the relationships between pairs of skills across academia to forecast how skills change over time. Based on our out-of-sample earnings prediction evaluation with *5-fold cross validation*, we also discover that differences in acquired skills help to explain the variance of graduates' earnings. Our results offer an approach that connects college education to future careers. These insights may enable educational policy and academic programs to adapt to the skill dynamics in the labor market. For example, information systems that bridge between higher education and workforce skill data may inform updates to course design that prepare students with the necessary skills for their desired careers.

In summary, this paper attempts to answer these following research questions:

- Q1. Can the granular workplace activities used by the Department of Labor to describe the US workforce also distinguish between different college majors and institutions?

- Q2. How do the DWAs taught in a curriculum or field of study evolve over time? Can the relationships between pairs of skills across all of academia help to predict the skill evolution?

- Q3. Do the differences in taught skills during higher education predict graduates' earnings? Similarly, do differences in taught skills within college majors correspond to earnings differences of recent graduates?

In the next section, we describe multiple datasets that enable us to answer aforementioned research questions. We then describe our methodology in detail, present our analysis and discuss its implications and potential weaknesses to conclude the paper.

## 2 Materials

**Open Syllabus Project Dataset** (https://opensyllabus.org (OSP)) is one of the largest corpora of syllabi in the world. As of October of 2019, it contains over eight million syllabi, collected from 5,381 colleges and universities, including over three million syllabi taught at 3,186 US institutions. OSP's fields-of-study classifier draws heavily from the Classification of Instructional Programs (CIP) taxonomy used by the National Center for Education Statistics to determine the academic field of study (*e.g.*, *Economics, Business, Computer Science*) best associated with each syllabus. It includes 62 fields of study. Each syllabus has a unique identifier and the text assignment data including a description of its content, a list of references and recommended readings, and course requirements (such as assignments and exams). Syllabi can be directly mapped to graduation and enrollment statistics from the US Department of Education's Integrated Postsecondary Education Data System (IPEDS). Syllabi are annotated with metadata including the institution, department, and academic year associated with the course. We extract and concatenate course titles, course descriptions and learning objectives from syllabi's textual data to create "course descriptions." More details can be found in SI Section 1 in S1 File. We limit the data from 2008 and 2017 (the ten most recent years in OSP), resulting in roughly 1.4 million syllabi representing college courses from 1,481 institutions. More about courses statistics per year and/or per field of study (FOS) can be found in S12 and S13 Figs in S1 File.

**O\*NET Detailed Work Activity (DWA) Taxonomy** (https://www.onetonline.org/help/online/dwa). O\*NET is designed and produced by the U.S. Department of Labor/Employment and Training Administration. The O\*NET database allows snapshots of the relationships between occupations and skills. It has 2070 DWAs (*e.g.*, *"develop methods of social or economic research.", "design integrated computer systems.", "design public or employee health programs."*) representing specific work activities performed across a small to moderate number of occupations within a job family. For example, the occupations with related activities to DWA *"design public or employee health programs."* include "Preventive Medicine Physicians", "Occupational Health and Safety Specialists", "Occupational Health and Safety Technicians", "Dietitians and Nutritionists", and "Dentists, General".

**Integrated Postsecondary Education Data System** (https://nces.ed.gov/ipeds/) (**IPEDS**) is the core postsecondary education data collection program of the U.S. Department of Education's National Center For Education Statistics (NCES). It annually collects information from all providers of postsecondary education, including public institutions, private nonprofit institutions, and private for-profit institutions, in fundamental areas such as enrollment, program completion and graduation rates. Providing data is required for any institution that applies for or participates in any Federal financial assistance program. IPEDS also includes a wide range of information about institution and institution groups, such as Degree-granting status, Institutional category, and Carnegie classifications. The Carnegie Classification, or more formally, the Carnegie Classification of Institutions of Higher Education (https://carnegieclassifications.iu.edu/), is a framework for categorizing all accredited, degree-granting institutions in the United States. It is designed to group colleges and universities based on their research activities.

**College Scorecard** (https://data.ed.gov/) is a U.S. Department of Education data initiative providing transparency and consumer information related to individual institutions of higher education and individual fields of study (*e.g.*, majors) within those institutions. College Scorecard provides information about post-college earnings including median earnings of graduates working and not enrolled after completing highest credential in their first and second years for the two graduation cohorts of years 2016 and 2017. We only use the first year earnings of

graduates. We process the data for Baccalaureate colleges and universities, and create the mapping between College Scorecard CIP code and OSP CIP code (the mapping can be found in this GitHub folder (https://github.com/HungChau/OSP-connect-higher-education/tree/main/cip_code_mapping)). As a result, we obtain 9007 earnings records for 832 institutions in 54 fields-of-study.

## 3 Methods and results

### 3.1 Modeling course syllabi with workplace skills

Are the workplace activities tracked by the US Department of Labor robust and effective to describe the knowledge in higher education? The O*NET database is produced by the US Bureau of Labor Statistics and details the labor market trends of workplace skills and activities by occupation. Specifically, detailed work activities (DWAs) are elements in the O*NET database that provide information about occupations' labor requirements. This data has been used to analyze several labor market dynamics including job polarization [11, 18] and the economic resilience of cities [3, 19]. Although O*NET relates occupations to skills in the workforce, similar data is not reported for educational programs even though many high-skilled workers obtain skills in college before entering the workforce.

We bridge this gap by detecting O*NET's detailed work activities from syllabus course descriptions. Each syllabus in the OSP data contains a description of the course content, a list of references and recommended readings, and course requirements, such as assignments and exams. Given a syllabus, we extract the course's title, description, and learning objectives from the text and concatenate them to form the *course descriptions* (details are in SI Section 1A in S1 File). We apply word embeddings [20] and document similarity techniques from natural language processing to represent each DWA and syllabus as continuous vectors distributed in the same pre-trained language embedding space. Language embedding models enable us to describe the semantic similarity between two textual documents or sentences; here, we compare syllabus course descriptions to DWAs. We choose pre-trained *fastText* word embeddings from [21], which is constructed from all Wikipedia pages in 2017, the UMBC webbase corpus, and the statmt.org news data. We choose these word embeddings because the semantic diversity of Wikipedia and news articles should capture the semantic diversity of topics taught across FOS. This model has been used in several applications [22–24], and achieves better performance than simple bag-of-words and TF-IDF [15]. We compute the *relationship* ($0 < = r_s(dwa) < = 1$) between a syllabus $s$ and a DWA by comparing their word embedding vector representations with soft cosine measure [25] (details are in SI Section 1B). As a result, syllabi are represented based on their relationships with the DWAs (called the DWA-based syllabus representation). We provide an example of the most and least prevalent DWAs detected for a political science syllabus at Harvard University in 2013 (see Fig 1A).

In addition to course descriptions, syllabi are annotated with metadata about where and when the course was taught. Metadata includes the institution, department/major/FOS, and academic year. OSP's field classifier is trained and tested on the IPEDS 2010 CIP taxonomy to determine the academic field (*i.e.*, FOS) best associated with each syllabus. This enables us to calculate the relationship between each pair of DWAs based on the co-occurrence of $dwa_1$ and $dwa_2$ in any set of course syllabi $S$; for example, the set of all syllabi within a given FOS, $sim_f(dwa_1, dwa_2)$ for $f \in FOS$, or across all of academia, $sim(dwa_1, dwa_2)$. We experiment with various semantic distance metrics to compute DWA relationships through syllabi including Jaccard similarity, Cosine similarity, Euclidean distance, and Manhattan distance (see SI Section 2 in S1 File). We find Jaccard similarity to be the most predictive and we present those results in the main text. It is worth noting that relationships between two DWAs can be

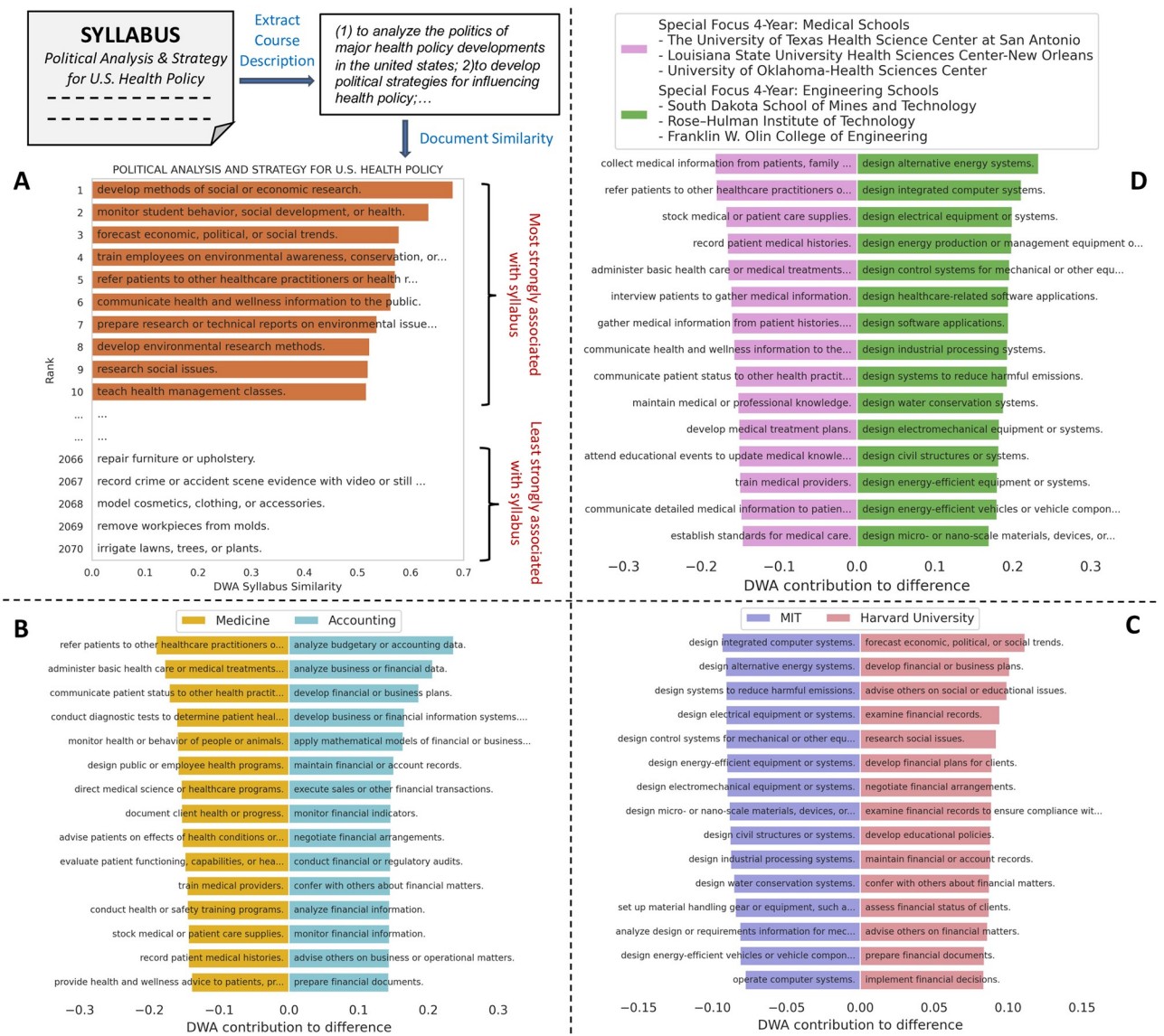

**Fig 1. The work activities inferred syllabi reveal key differences among universities and fields of study.** (A) An example political science syllabus from Harvard University and the activities that are most and least strongly associated with its course description. DWA-syllabus similarity scores range from *0* (not detected) to *1* (strongly detected). (B) The DWAs that most significantly distinguish Accounting syllabi from Medicine syllabi. (C) The DWAs that most strongly separate MIT syllabi from Harvard syllabi. (D) The DWAs that most strongly separate Special Focus 4-Year Medical Schools syllabi from Engineering Schools syllabi. More examples can be found in S1-S4 Figs in S1 File.

directly computed by measuring the cosine similarity of their embedding vectors. However, this approach measuring a static relationship between DWAs fails to distinguish the dynamics of how one DWA relates to another locally (i.e., within a FOS or a university) and globally (i.e., across all of academia) overtime, which will be discussed in Section Predicting the change in taught skills. For example, social skills and computer programming skills may be semantically different but co-taught as complementary skills across syllabi (e.g., computational social science, social network analysis, or econometrics).

The syllabus-DWA relationships ($r_s(dwa)$) also enable us to model a FOS $f$ and a university $u$ in terms of their relationship to each of the DWAs according to, respectively,

$$r_f(dwa) = \frac{1}{|S_f|} \sum_{s \in S_f} r_s(dwa) \qquad \text{and} \qquad r_u(dwa) = \frac{\sum\limits_{f \in FOS} \sum\limits_{s \in S_{f,u}} \alpha_{f,u} \cdot r_s(dwa)}{\sum\limits_{f \in FOS} \alpha_{f,u} \cdot |S_{f,u}|}. \qquad (1)$$

These relevance scores are a measure of how strongly the skill (i.e., *dwa*) is represented in a field or university. While $r_f(dwa)$ (the relevance score of the *dwa* to FOS $f$) is the average over the similarity scores of that DWA across $s \in S_f$, $r_u(dwa)$ (the relevance score of the *dwa* to university $u$) is the mean similarity score of that DWA across syllabi weighted by the estimated graduation rates ($\alpha_{f,u}$) of the syllabus's field of study at that university. In the absence of course enrollment data, we use graduation rates for each FOS at each university to approximate the number of students who learn from each syllabus. $S_f$ represents all of the syllabi within a given FOS $f$, and $S_{f,u}$ represents all of the syllabi within a given FOS at a university $u$.

These tools enable us to compare pairs of syllabi, FOS, or universities based on their most common DWAs. We publish the DWA similarities by different metrics, DWA scores for each FOS and for each university by year from 2008 to 2017 in a Github repository (https://github.com/HungChau/OSP-connect-higher-education). Specifically, we compare entities of the same type (*e.g.*, one FOS to another) by subtracting its DWA vector representation from the other's and rank the resulting vector in descending order. We visualize the top 15 DWAs of each entity that contribute most to the difference of the pair in Fig 1B–1D. For example, the DWAs "refer patients to other healthcare practitioners or health resources" and "administer basic health care or medical treatments" most strongly distinguish Medicine from Accounting, while "analyze budgetary or accounting data" and "analyze business or financial data" identify Accounting from Medicine (see Fig 1B). Similarly, we compare pairs of universities based on their taught DWAs. As an example, "design integrated computer systems" and "design alternative energy systems" most strongly distinguish Massachusetts Institute of Technology (MIT) from Harvard University, while "forecast economic, political, or social trends" and "develop financial or business plans" more strongly identify Harvard from MIT (see Fig 1C). These results match our intuition as MIT is the world-leading engineering university and Harvard is in the top ten universities in each social science area according to U.S. News rankings. Building on this, we can group universities based on their Carnegie classification to identify the major differences in taught DWAs. We compare Medical Schools to Engineering Schools in Fig 1D. More examples can be found in S1-S4 Figs in S1 File.

## 3.2 Identifying Field-of-Study and university clusters

Do DWAs capture the focal knowledge offered by an academic field or a university? To further compare education among FOS, we use agglomerative hierarchical clustering on DWA-based vector representations of each FOS. Hierarchical clustering [16] produces a nested sequence of clusters like a tree (also called a dendrogram). Agglomerative clustering builds the dendrogram from the bottom level, and merges the most similar (or nearest) pair of clusters at each level to go one level up. Hierarchical clustering can take any form of distance or similarity function, and the hierarchy of clusters enables us to explore clusters at any level of detail without the need of picking a number of topics $k$ as would be the case with K-means clustering. Pairs of FOS are similar if they are associated with similar types of work activities. For instance, *Accounting* is clustered together with *Business* and *Marketing*; *Medicine* is clustered together with *Nursing*, *Nutrition*, *Health Technician*, *Dentistry* and *Veterinary Medicine*; the STEM

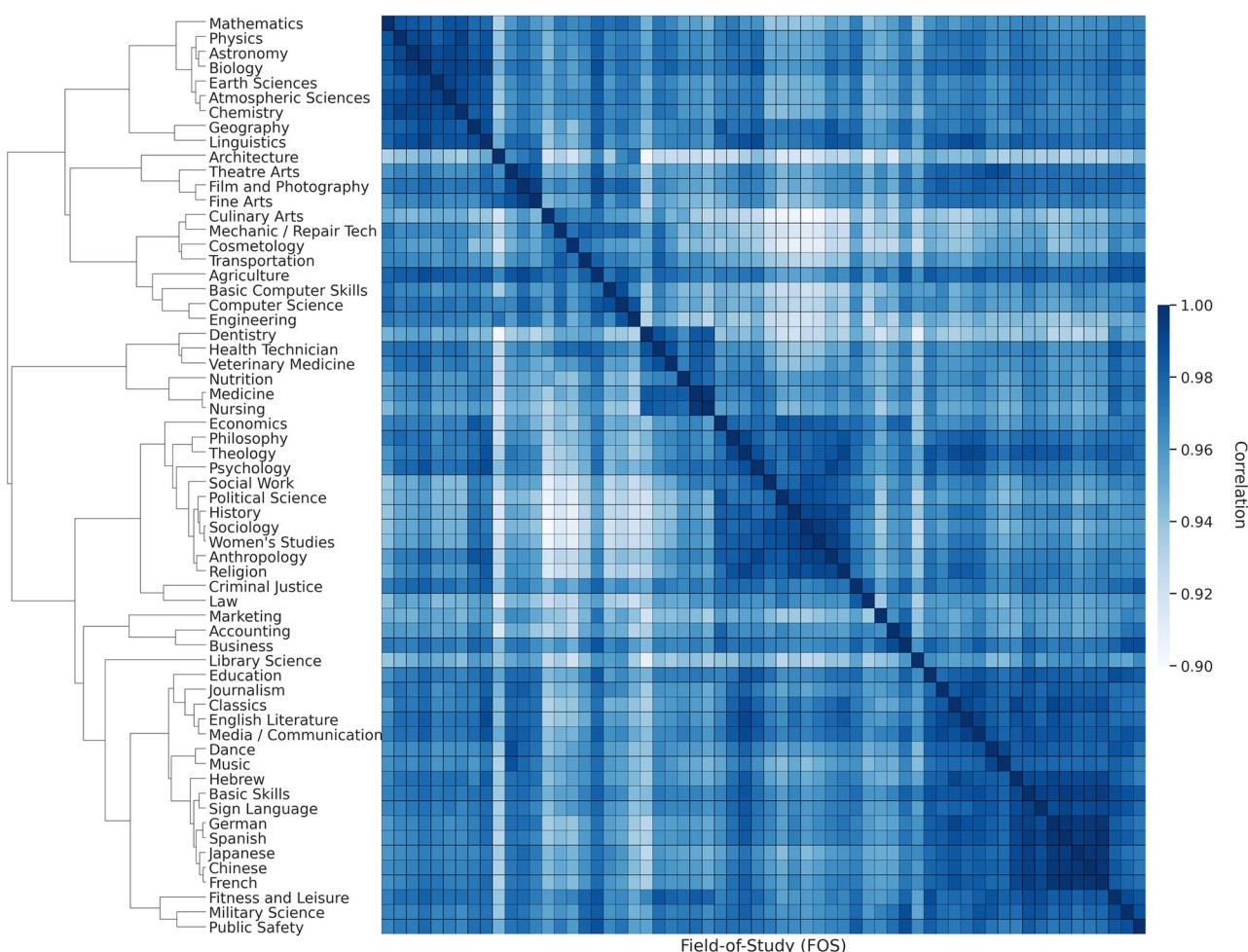

**Fig 2. The similarity of FOS based on the prevalence of DWAs in syllabi from within those fields.** The dendrogram and heatmap show similar FOS clustered together based on their DWA-vector representations.

cluster includes *Mathematics*, *Physics*, *Astronomy*, *Biology*, *Earth Sciences*, *Atmospheric Sciences* and *Chemistry*; and the Social Science cluster includes *Social Work*, *Political Science*, *History*, *Sociology*, *Women Studies*, *Anthropology* and *Religion* (see Fig 2).

Similarly, we compare all US universities in our data set using agglomerative hierarchical clustering performed on the *weighted* DWA-based vector representation of each institution in Fig 3. We see that similar universities are clustered together. For example, *The University of Texas Medical Branch*, *The University of Texas Health Science Center*, and *Oregon Health and Science University* are clustered together. Although our dataset contains a large number of universities, we select a subset of Ivy Plus universities and universities from various IPEDS Carnegie Classifications to visualize in Fig 3. We filter out universities that have less than 100 syllabi or were missing syllabi in any year from 2008 to 2017. Carnegie classifications are mostly recovered by the clusters (see colors in Fig 3). Additionally, engineering schools like *California Institute of Technology*, *Massachusetts Institute of Technology*, and *Carnegie Mellon University*, are clustered together. Similarly, liberal arts schools including *Cornell University*, *Harvard University*, and *University of Pennsylvania* are clustered together.

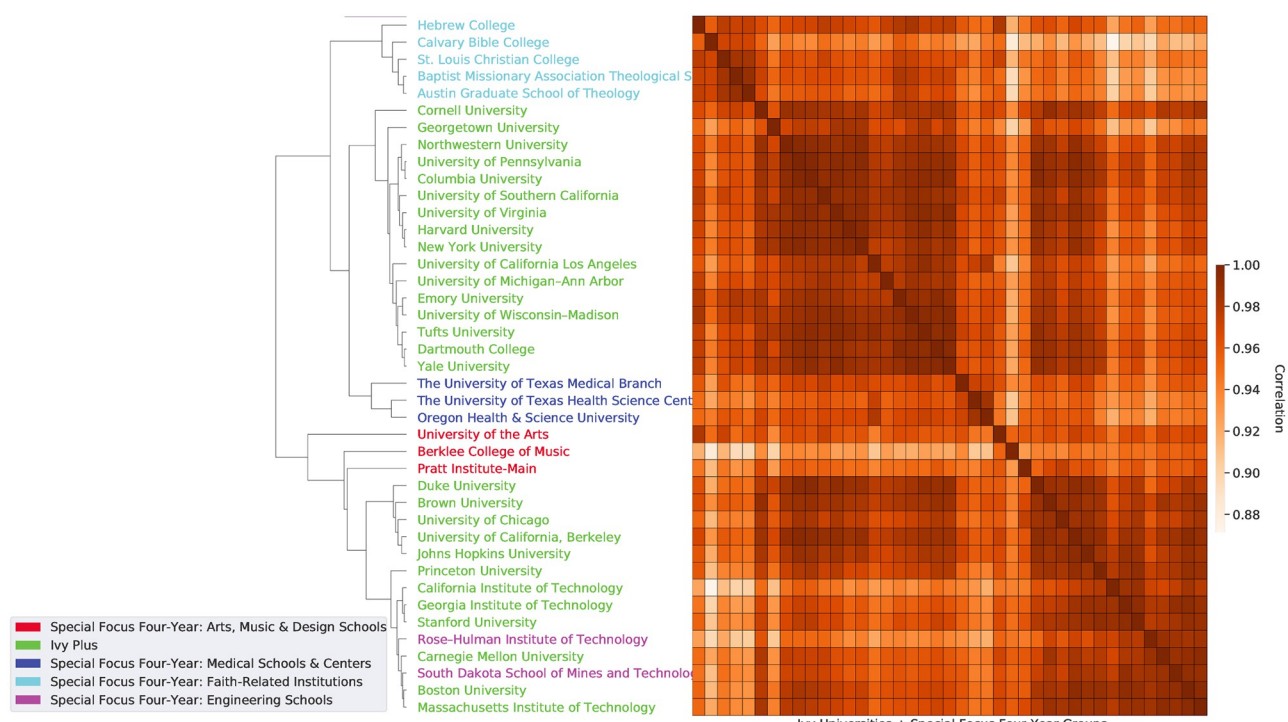

**Fig 3. The similarity of universities based on the graduation-weighted prevalence of DWAs offered in their course syllabi.** The dendrogram and heatmap reveals the hierarchical clustering of the Ivy Plus group and Special Focus Four-Year groups from the Carnegie Classification 2018 based on DWA vector representations.

## 3.3 Predicting the change in taught skills

How do the DWAs taught in a field of study evolve over time? In particular, which new skills or topics will emerge in a field's syllabi? Forecasting these educational trends enables proactive course design by educators and could inform educational incentives from policy makers. Here, we use the principle of relatedness [17] to hypothesize that DWAs that occur together across all of higher education are more likely to be co-taught within a given FOS in the future. If correct, then modeling the relationships between pairs of DWAs across all of academia should forecast the introduction of new topics within a FOS even if that topic has not been part of that FOS historically. As an illustrative example, although largely absent from Economics syllabi today, machine learning may become more common in Economics because Economics already teaches linear regression which is commonly taught as an example of machine learning in Computer Science courses. As a more specific example from our data, DWAs that relate to machine learning, such as "analyze website or related online data to track trends or usage" may become more prevalent in Economics syllabi moving forward (e.g., in studies of online job postings [12, 26]).

We test our hypothesis using OSP data to predict which DWAs become important in a FOS ($f$). We use the relevance scores ($r_f(dwa)$) calculated from the syllabi of each FOS in two different years (i.e., 2008 and 2017). We recast this problem as predicting the score difference ($\Delta r$) of a DWA between the two years:

$$\Delta r_{dwa,f} = r_f^{2017}(dwa) - r_f^{2008}(dwa) \tag{2}$$

We also perform classification analysis for predicting DWAs becoming important in future, which can be found in SI Section 3B in S1 File. We run several ordinary least squares (OLS) regressions to predict $\Delta r_{dwa,f}$ using the relevance scores of the $dwa$ to FOS $f$ ($r_f(dwa)$) and various models of inter-DWA relationships (described in Section Modeling course syllabi with workplace skills). As a baseline, we first consider Model 1 using only the current relevance scores of DWA within each FOS with FOS fixed effects (denoted $\lambda_f$) according to

$$\Delta r_{dwa,f} = \beta_0 + \beta_1 r_f^{2008}(dwa) + \lambda_f. \tag{3}$$

Next, we additionally include a variable representing the co-occurrence of DWAs across syllabi within a FOS (denoted $R_f$) to create Model 2

$$\Delta r_{dwa,f} = \beta_0 + \beta_1 r_f^{2008}(dwa) + \beta_2 \underbrace{\left( \frac{\sum_{dwa' \in DWA} sim_f(dwa, dwa') r_f^{2008}(dwa')}{|DWA|} \right)}_{R_f} + \lambda_f \tag{4}$$

and yet another similar Model 3 using DWA pair co-occurrences across syllabi from every FOS (denoted $R$)

$$\Delta r_{dwa,f} = \beta_0 + \beta_1 r_f^{2008}(dwa) + \beta_2 \underbrace{\left( \frac{\sum_{dwa' \in DWA} sim(dwa, dwa') r_f^{2008}(dwa')}{|DWA|} \right)}_{R} + \lambda_f. \tag{5}$$

Model 4 includes an interaction term between DWA's relevance score within a FOS (*i.e.*, $R_f$) and DWA pair co-occurrences within that FOS according to

$$\Delta r_{dwa,f} = \beta_0 + \beta_1 r_f^{2008}(dwa) + \beta_2 R_f + \beta_3(r_f^{2008}(dwa) * R_f) + \lambda_f \tag{6}$$

and, in Model 5, using DWA pair co-occurrence across all FOS

$$\Delta r_{dwa,f} = \beta_0 + \beta_1 r_f^{2008}(dwa) + \beta_2 R + \beta_3(r_f^{2008}(dwa) \cdot R) + \lambda_f \tag{7}$$

As robustness checks, we run Models 2, 3, 4 & 5 with the two different methods and four distance metrics aforementioned in Section Modeling course syllabi with workplace skills for computing the DWA relationships. Although we could compare DWA pairs based solely on their semantic similarity using their word embedding vectors, this approach would miss DWA pairs that capture complementary topics. For example, Models 2 and 3 would be identical to Models 4 and 5, respectively. The results (see SI Section 3A in S1 File) show that modeling DWA relationships based on their co-occurrence in syllabi with Jaccard similarity yields the best performances across all the models involving inter-DWA relationships. We discuss these results in the main text.

We compare model performance using root mean squared error (RMSE) with 5-fold cross validation in Fig 4 (R-squared metric is reported in S11A Fig in S1 File). First, including variables representing DWA relationships decreases RMSE (*i.e.*, Model 2 ($R^2 = 0.231$) & Model 3 ($R^2 = 0.239$) are statistically significantly better than Model 1 ($R^2 = 0.191$)). Second, measuring DWA co-occurrences across all of academia (i.e., using $R$) instead of only within a single FOS

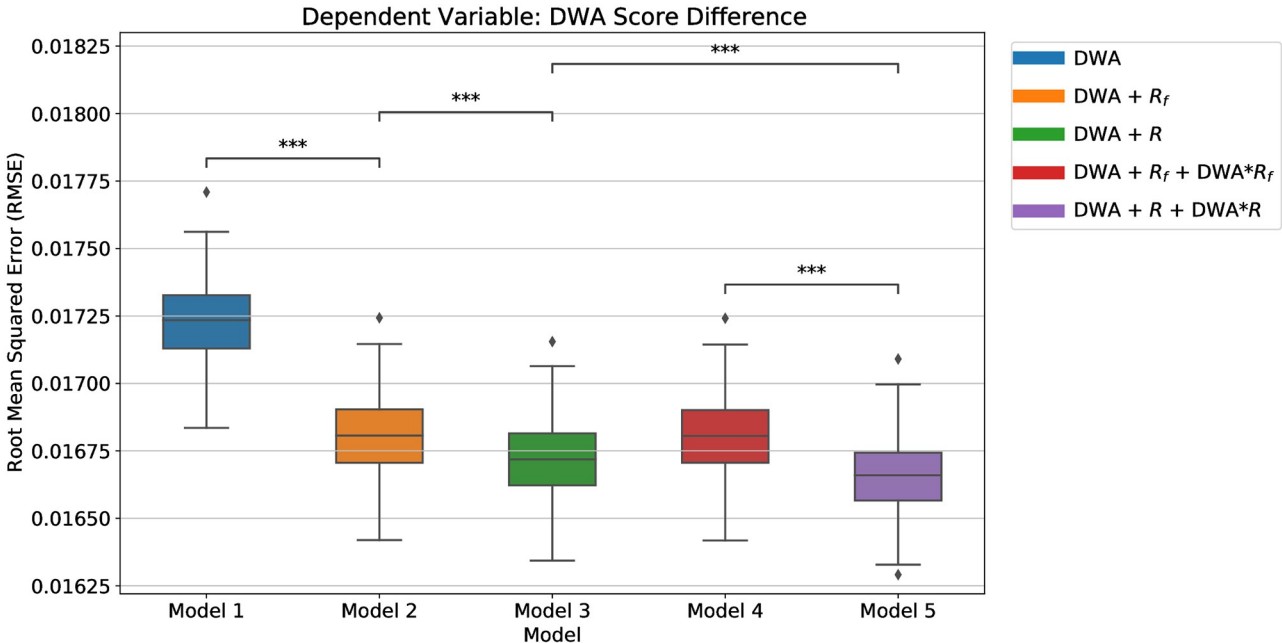

**Fig 4. Workplace activities detected from syllabi predicting teaching dynamics within a field of study.** We perform 5-fold cross validation and repeat 40 times (i.e., 200 trials in total) for each model and measure RMSE by the resulting model applied to the test set. Asterisks indicate the statistically significant difference between two models' performances with Bonferroni correction. Predicting the importance of DWAs changing in nine years (2008 vs. 2017). As a baseline, model 1 only considers the current DWA score and FOS fixed effects. The other models consider the relationships between DWAs, how they interact with each other to predict how they may change in future.

(i.e., using $R_f$) improves model predictions. Specifically, Model 3 ($R^2 = 0.239$) outperforms Model 2 ($R^2 = 0.231$) and Model 5 ($R^2 = 0.244$) outperforms Model 4 ($R^2 = 0.231$).

These results suggest that FOS educational trends within a FOS correspond to global educational trends across all of academia. In particular, this evidence supports our hypothesis that DWAs tend to be co-taught more within a given FOS if they are bundled together across all of higher education. (*e.g.*, Computer Science may increasingly teach "analyze green technology design requirements" since it is commonly taught with "identify information technology project resource requirements" in other FOS including Engineering). Although Model 4 does not outperform Model 2, including the interactions between current DWA relevance scores and the average of the proximity of *global* DWA relationships does yield a significant improvement (*i.e.*, Model 5 outperforms Model 3). In conclusion, the best performing model is Model 5 which leverages the information about the current score of the DWA, their relationships with other DWAs across academia, and the interaction of these two variables. Model 5 improves 3.3 percent (27.5 percent) in terms of RMSE (R-squared) over Model 1, which only uses the *2008* DWA relevance scores. Therefore, we train Model 5 using the entire data, and use it to predict the relevance scores of DWAs in a FOS nine years later. Table 1 shows some examples of DWAs that became important within a FOS —in terms of ranking DWAs—in nine years. The full list of DWAs that are predicted to increase their ranks by at least five units and ranked in the top 50 in 9 years can be found in the aforementioned Github repository.

### 3.4 Predicting graduate earnings

Do detected DWAs predict the variation in graduates' earnings? Most—if not all—educational programs aim to provide students with the skills and abilities to successfully enter the

**Table 1. Examples of DWAs that are predicted to increase their ranks in 9 years in particular fields.** We only select DWAs that are ranked in top 50 in future. The full list of predicted DWAs can be found in the same Github folder.

| Field-of-Study | Detailed Work Activity | Rank (2017) | Rank (2026) |
|---|---|---|---|
| Computer Science | analyze green technology design requirements. | 40 | 33 |
| | apply information technology to solve business or other applied problems. | 46 | 40 |
| Economics | evaluate plans or specifications to determine technological or environmental implications. | 37 | 27 |
| | develop marketing plans or strategies for environmental initiatives. | 58 | 50 |
| Journalism | gather information about work conditions or locations. | 37 | 24 |
| | prepare scientific or technical reports or presentations. | 48 | 42 |
| Medicine | develop healthcare quality and safety procedures. | 28 | 23 |
| | operate laboratory equipment to analyze medical samples. | 65 | 50 |
| Physics | develop procedures for data entry or processing. | 43 | 33 |
| | develop performance metrics or standards related to information technology. | 41 | 34 |

workforce (*e.g.*, to gain employment and maximize earnings). Most empirical work relies on coarse labor distinctions such as college major and institutional information (e.g., school brands) to correlate to graduate earnings [7, 9, 27, 28], but none have provided insights into the skills students learn that could contribute to their future earnings. Our analysis of DWAs in university course syllabi provides the first data set connecting taught skills to students' earnings after graduation. We collect earnings of graduates from the College Scorecard earnings data from the U.S. Department of Education. Though large, the OSP course syllabus data is not distributed evenly across fields-of-study and institutions. Some fields and institutions have much less course syllabi. Thus, to sufficiently estimate work activities taught in a FOS at a university, we limit earnings records for FOS (in an institute) that have at least 10 course syllabi; and perform Kolmogorov-Smirnov statistical test to make sure the remaining earnings records representative for the entire population of the field at the institute (more details on the selection process and criteria are in SI Section 4 in S1 File). We build several OLS regression models to predict *average* graduate earnings across FOS (*f*) at a university (*u*) based on the relevance scores of the DWAs across fields (*DWA*) and within field (*FOS*DWA*), FOS fixed effects (*FOS*), school brands (i.e., school ranks (Historical U.S. News and World report rankings are compiled by Andy Reiter and available at https://andyreiter.com/datasets/) if available) fixed effects (RANK), and geography fix effects (*GEO*). Due to the limited availability of earnings data, we use groups of 10 ranks (i.e., 1–10, 10–20) for national universities and 15 ranks (i.e., 1–15, 15–30) for liberal arts colleges. For geographical features, we group universities together based on their divisions (U.S. Geographic Levels are available at https://www.census.gov/programs-surveys/economic-census/guidance-geographies/levels.html) (e.g., New England Division, West North Central Division). These groups are represented using indicator variables in the regression analyses.

To avoid model over-fitting, we perform 5-fold cross validation and LASSO feature selection on the models that include DWA features. LASSO [29] is one of the most popular methods for feature selection; it minimizes the residual sum of squares subject to the sum of the absolute value of coefficients being less than a constant. This constraint tends to "regularize" large models by producing some 0 coefficients when variables are co-linear. In other words, the penalty factor determines how many features are retained; using cross-validation to choose the penalty factor helps assure that the model will generalize well to future data samples. As a

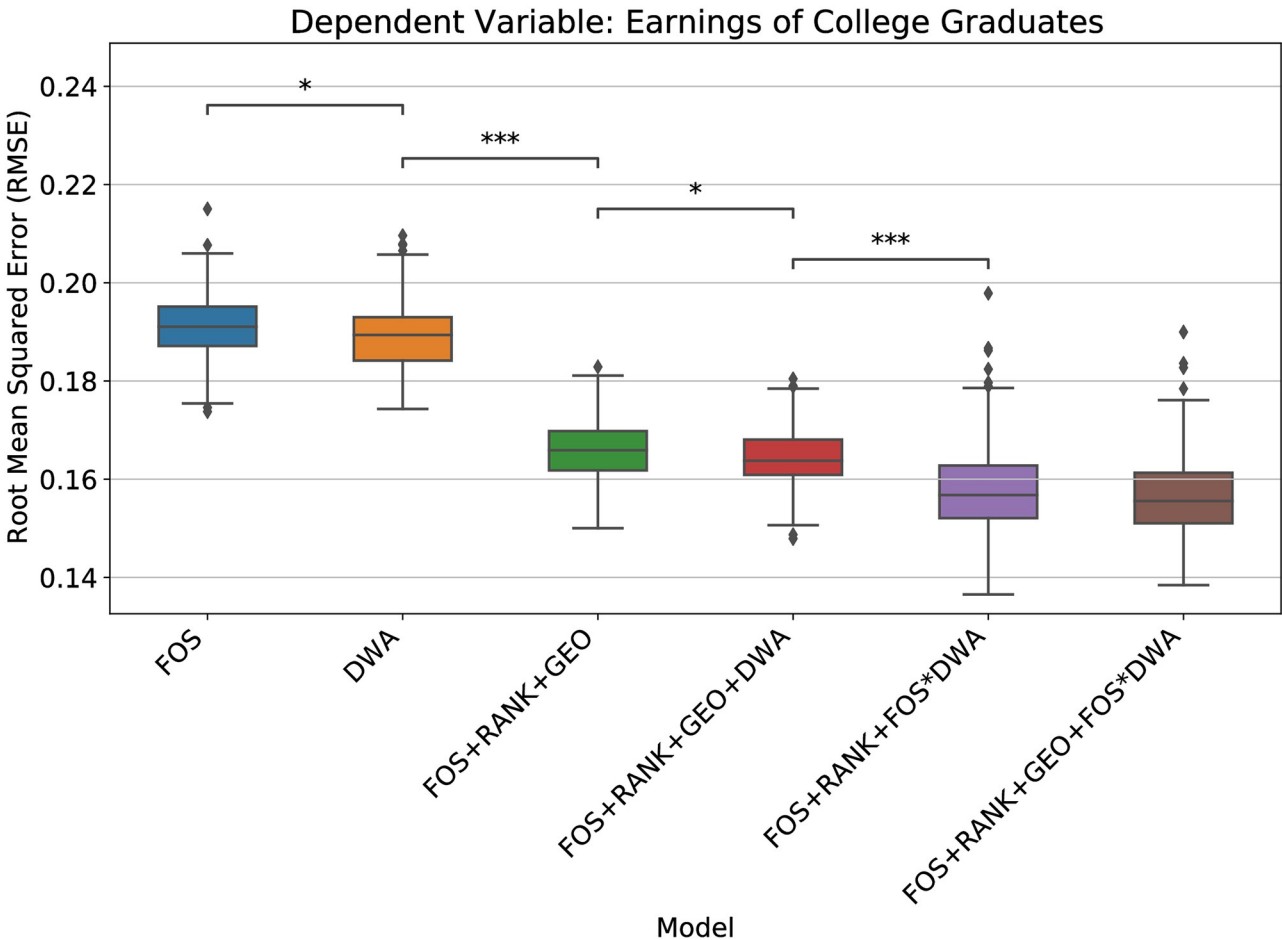

**Fig 5. Workplace activities detected from syllabi predicting median first-year earnings of college graduates across fields of study.** We perform 5-fold cross validation and repeat 40 times (i.e., 200 trials in total) for each model and measure RMSE by the resulting model applied to the test set. Asterisks indicate the statistically significant difference between two models' performances with Bonferroni correction. As a baseline, we consider the FOS, school ranking, and geographic fixed effects to predict earnings.

result, we find that DWAs improve predictions of graduate incomes (see Fig 5 for *RMSE* metric and S11B Fig in S1 File for *R-squared* metric according to 5-fold cross validation). Including DWAs improves predictions of earnings compared to FOS fixed effects (*i.e.*, smaller RMSE). Also, $R^2 = 0.684$ of the *DWA* model is significantly better than that of *FOS* model $R^2 = 0.677$. Controlling for university rankings and geography further improves the *FOS* model (i.e., *FOS+RANK+GEO* ($R^2 = 0.757$) model is significantly better than *FOS* ($R^2 = 0.677$) model). But combining DWA variables with RANK and GEO variables and FOS fixed effects yields even further improvement (*FOS+RANK+GEO+DWA* model ($R^2 = 0.761$) is statistically significantly better than that of *FOS+RANK+GEO* model). This evidence suggests that some of the information about graduate earnings represented in university rankings is also encoded the DWA variables (e.g., a LASSO regression model containing DWA variables accounts for 48% of the variation in college rankings; year and FOS fixed effects account for 7.9%). Finally, the best model (*FOS+RANK+FOS*DWA*) is found when we allow DWA variables to interact with FOS fixed effects which suggests that different DWAs correspond to earnings variation in different FOS ($R^2 = 0.779$). The geographic variables also help to improve the best model's performance but not significant ($R^2 = 0.782$).

### 3.5 Within Field-of-Study skill variation and the earnings of recent college graduates

Do differences in taught skills within college majors correspond to earnings differences of recent graduates? To study how DWAs relate to earnings of graduates of a specific field of study, we perform separate regression analyses for each FOS with at least 100 institution-year observations. We employ LASSO feature selection for DWAs and report model performance using 40 independent trials of 5-fold cross-validation to mitigate over-fitting. The remaining DWAs are used to predict earnings. As can be seen from Fig 6, the *DWA+GEO* models perform significantly better than the baseline *GEO* models in terms of RMSE. Due to the limited earnings data within FOS to perform cross validation, the school ranking is omitted; the baseline models only include geographic variables (*GEO*). We obtain similar performance when alternatively using the model variance explained ($R^2$) (see S11C Fig in S1 File). This result again shows that the DWAs complement the FOS information by increasing the share of the earnings explained by the model and improving the model's predictions. However, *DWA +GEO* model performance varies across FOS. For example, the *DWA+GEO* model improves 27.2% RMSE over the *GEO* model for *Business* compared to a more modest improvement of 4.2% for *Psychology*. Although O∗NET DWAs improve predictions in general, this varied performance across FOS could be because DWAs represent key skills and activities better in some FOS than in others. Nevertheless, our methodology shows that using granular workplace skills helps to identify important features contributing to earnings of graduates beyond course educational and labor categories.

Identifying DWAs that correspond to increased earnings after graduation could inform students' course selection based on the demand for skills in the labor market. To demonstrate this, we analyze the regression of FOS *Business* as an example. After performing 5-fold cross validation on the model determined by LASSO feature selection, there are 57 DWAs remaining. Based on our statistical regression analysis, the 57 DWA features are able to explain 69.2% of the variance of the earnings in *Business*. Among those, 10 DWAs have significant coefficients with the *p-values* below 0.05. DWAs "*complete documentation required by programs or regulations,*" "*evaluate program effectiveness,*" and "*advise others on career or personal development*" are positively associated with earnings while "*conduct health or safety training programs*" is negatively associated with earnings (regression coefficients estimated with $p_{value} < 0.01$ in each case). The list of DWAs have significant coefficients for all the 10 FOS can be found in S2 Table in S1 File. The full list of all the selected DWAs including the coefficients and statistics can be found in this GitHub folder (https://github.com/HungChau/OSP-connect-higher-education/tree/main/selected_DWAs).

## 4 Discussion

Knowledge, skills, and abilities shape workers' careers, and so, quantifying their sources may impact workforce development and our understanding of the labor market. Largely, higher education is a source of skill acquisition for many middle and high-skilled jobs in America. However, there is a disconnect between work and learning in the US; higher education can fail to meet the skill demands of the labor market thus creating "skill gaps" across the country. A labor market information system where work skills are shared across entities, connecting education to work, could help students know what skills they need, educators know what skills to instruct for, employers know what skills workers have, and policy makers more effectively impact workforce development. This study demonstrates a methodology to bridge material taught in U.S. colleges and universities with the detailed work activities (DWAs) used by the Department of Labor to describe the US workforce. This creates new opportunities to track

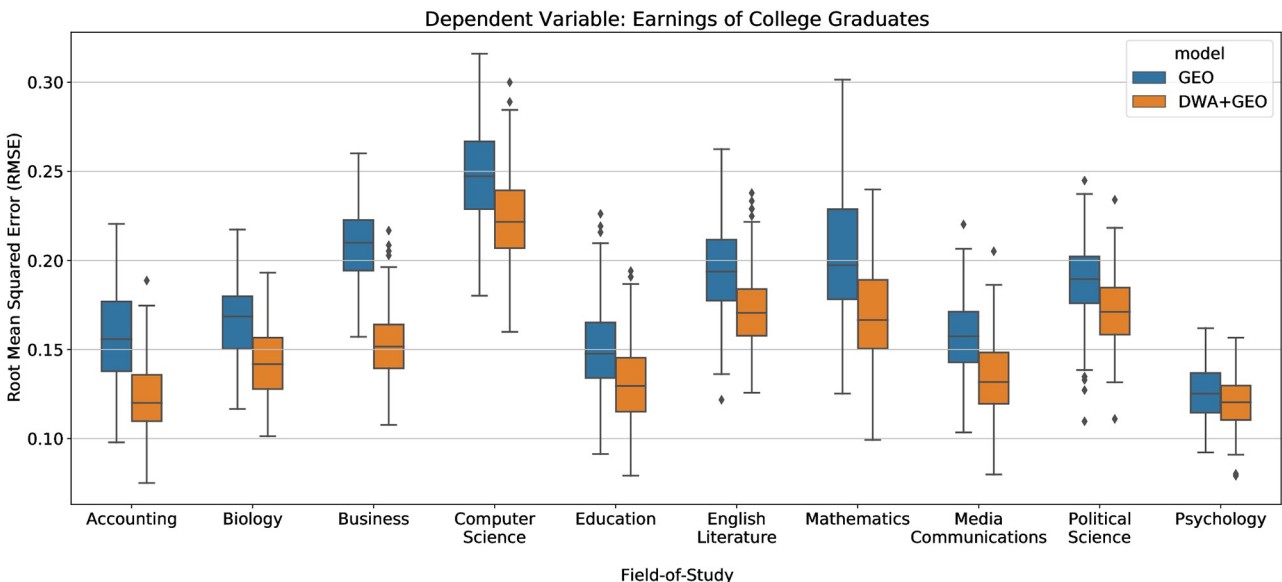

**Fig 6. Workplace activities detected from syllabi predicting median first-year earnings of college graduates within a field of study.** We perform 5-fold cross validation and repeat 40 times (i.e., 200 trials in total) for each model and measure RMSE by the resulting model applied to the test set. The baseline *GEO* model only includes geographic variables. The performances of the *DWA+GEO* models are statistically significantly better than the *GEO* models with the *p-values* $< 0.05$ for all of the reported FOS (the school ranking is omitted due to the limited earnings data).

changes in the evolution of higher education and workforce development; for example, the emergence of DWAs within the syllabi of a field of study (FOS), or major, corresponds to the co-occurrence of DWA pairs across all of academia (see Fig 4). As an illustrative example, discussions of green technology design requirements may become more prominent in Computer Science programs because they go hand-in-hand with information technology project resource requirements, commonly taught in courses across academia. Educators, educational policy, and course recommendation systems could use these insights to design educational programs and to advise students towards the classes offering the experience that will be most valuable for their career goals. Following our example, proactive curriculum design might include green technology topics to prepare students for jobs in Computer Science.

However, it is likely not the case that every FOS will teach every skill or ability, in part, because labor market incentives for specific DWAs vary by industry, region, and employer. Thus, insights into the course topics that correspond to increased, or decreased, earnings after graduation (see Fig 6 for example) may increase the relevance of an educational program or policy and increase students' success when they enter the workforce. For example, academic programs might grow to include new high-demand skills while decreasing emphasis on outdated topics. Such insights could inform *goal*-based learning [30] in course recommendation systems while improving explanations of recommendations. Increasingly-personalized course recommendations can identify relevant topics based on students' predefined goals (*e.g.*, maximizing job earnings). For example, recommending *Business* courses that include "*complete documentation required by programs or regulations*" work activities might proactively prepare today's students to meet the growing demand for Business Analytics in the labor market.

### 4.1 This study has a few limitations

This study demonstrates how novel syllabus data and natural language processing (NLP) techniques can connect labor market data to higher education by predicting the change in taught

skills within a FOS and linking DWAs to graduate earnings. Future work might build on our study by analyzing the causal implications of skill-level adjustments to course content. In particular, our study's approach is unable to address selection bias when students choose a university in which to enroll. But future work may study natural experiments that overcome this barrier. Potential examples include the hiring, firing, or retirement of new faculty, the creation of a new school or department, the emergence of a large employer (*e.g.*, resulting from new tax credit), or large donations focused on specific learning outcomes. For example, future work might augment our analysis of graduate's recent earnings with other career outcome measures. Our analysis of the College Scorecard earnings data is limited to only two graduation cohorts and similar Post-Secondary Employment Outcomes data is limited to only a few institutions. Furthermore, we only consider earnings one year after graduation, which may not capture the full career trajectory [31]. However, future analysis involving workers' resumes will enable direct connections between workers' educational foundations during college and their career dynamics (*e.g.*, worker adaptability, tenure, and mobility) in addition to earnings. Similarly, job postings analysis might compare employer demands to the DWAs detected in our study thus identifying the most or least adaptive educational programs (*e.g.*, [12]). Future research along this dimension will offer new insights into the sources and sinks of the high-skilled workers that shape job polarization [11] and urbanization today [4, 19].

We have demonstrated, using mean cohort level graduate earnings, that there is already detectable variation in earnings based on skills taught in courses offered. Our approach has focused on outcomes for groups of graduates (e.g., by major or university). Future work with alternative data might investigate variations in labor market outcomes for individuals. For example, students studying the same major could take different courses offered, thus learning different skills. Whether the course selection by individual students leads to different occupations and different earnings, and how much learned skills could explain individual career variation are interesting questions left to be discovered. One challenge in undertaking such research is the availability and accessibility of this type of datasets at scale due to privacy concerns. Further, our analyses focused on students with bachelor's degrees, but future work might study the skills of graduate education or the undergraduate education that lead to graduate school admission.

Our study relied on simple off-the-shelf techniques in combination with novel data sources, but future work might expand our methods with more sophisticated approaches. For example, this study used pre-trained *static* word embeddings and standard document similarity techniques to detect work activities from syllabi, but more complex NLP techniques could yield further insights. Static word embeddings are a powerful tool for capturing syntactic and semantic regularities in language, but each word is represented by a single vector regardless of context. That is, all senses of a polysemous word have to share the same representation. *Contextualized* word representations, such as Transformer-based embeddings, overcome those issues and have yielded significant improvements on many NLP tasks. Additionally, our study relies on the O*NET taxonomy used by US Department of Labor to describe labor market trends. These granular DWAs reveal core differences between courses, fields and universities. For example, DWA relevance scores improved predictions of graduate earnings within many fields of study, but not all. This suggests that "skill" differences may impact the effectiveness of college education (in terms of earnings) but O*NET DWAs may not be the most precise taxonomy to describe the granular level of knowledge expressed in courses. This is in part because O*NET data is not designed to describe higher education, but to describe workers. There is no standard knowledge base describing more granular concepts and skills in higher education and the labor market. This highlights an urgent need for future educational research that builds a knowledge base that could standardize and advance insights into how educational

foundations shape workforce development and the skills of workers. With the advances of text mining methods, one could extract skills described in course syllabi and job postings, and align those skills to connect educational contents with the demands of the labor market. There are some existing job skill taxonomies to describe job postings' requirements such as BG's or LinkedIn's proprietary skill taxonomies. Börner et al. (2018) analyze course syllabi and BG's job postings focusing on areas of Data Science and Data Engineering. They use BG's skill taxonomy instead of the one used by the U.S. Bureau of Labor Statistics to analyze skill discrepancies between research, education and jobs. Modeling job postings with NLP techniques has also been shown to be useful in understanding wage premia [32]. Although our study focuses on the work side of job seeking, we acknowledge that the demand from the employer side is also important to understand the holistic picture from skill offerings in higher education to skill demands in the labor market; which could benefit many applications such as identifying potential curricular gaps or recommending courses to meet jobs' requirements.

Increasingly, researchers and policy makers use workers' skills and abilities to describe labor market outcomes in addition to workers' educational attainment based on their occupation [5]. But, similar data and methods are only just being developed and applied to workforce development and, in particular, to higher education. This study offers an approach and a methodology to connect higher education to workplace skills thus enabling new strategies for course recommendation, curriculum design, and education policy that prepare students to meet their career goals.

## Supporting information

**S1 File.**
(PDF)

## Acknowledgments

We thank Erik Brynjolfsson, Seth Benzell, Daniel Rock, Nabeel Gillani, and Peter Brusilovsky for their feedback throughout this project.

## Author Contributions

**Conceptualization:** Sarah H. Bana, Morgan R. Frank.

**Data curation:** Hung Chau, Sarah H. Bana, Baptiste Bouvier, Morgan R. Frank.

**Formal analysis:** Hung Chau, Sarah H. Bana, Baptiste Bouvier.

**Funding acquisition:** Sarah H. Bana, Morgan R. Frank.

**Investigation:** Hung Chau, Sarah H. Bana, Morgan R. Frank.

**Methodology:** Hung Chau, Baptiste Bouvier, Morgan R. Frank.

**Project administration:** Sarah H. Bana, Morgan R. Frank.

**Resources:** Morgan R. Frank.

**Supervision:** Sarah H. Bana, Morgan R. Frank.

**Visualization:** Hung Chau.

**Writing – original draft:** Hung Chau, Sarah H. Bana, Morgan R. Frank.

**Writing – review & editing:** Hung Chau, Sarah H. Bana, Morgan R. Frank.

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
