## [Decision Letter · Decision Letter 0]

4 Nov 2022

PONE-D-22-21083Connecting Higher Education to Workplace Activities and EarningsPLOS ONE

Dear Dr. Frank,

Thank you for submitting your manuscript to PLOS ONE. After careful consideration, we feel that it has merit but does not fully meet PLOS ONE’s publication criteria as it currently stands. Therefore, we invite you to submit a revised version of the manuscript that addresses the points raised during the review process.

The reviewer and I see value in this paper, given the novelty that it brings to the literature on skills acquisition in higher education and how they are related to the skills demanded in the labor market. I agree with the reviewer that this topic is very interesting and important.

After carefully reading the paper, I believe that the current version of the paper suffers from several limitations and needs some additional work. Here are my main comments on this version of your article.

The abstract should be re-written following the list of results found in the paper. First, relationships between syllabi and DWA; second, prediction of the evolutions of skills; third, the relationship between DWA associated with syllabi and earnings.  

Introduction. I agree with reviewer 1 that the introduction of the paper is not very effective. You start very far from your empirical exercises- even mentioning America’s position in the global economy. Then, you pose questions to which you provide no answer: “ what educational foundations best enable students to achieve their career goals?” You look at entry wage, not careers. Another example:  “How do academic majors change their curriculum over time?” while you describe some changes, do not explain how. And so on. Please, be more focused and effective in introducing your empirical study; after a succinct (3-4 lines) about the importance of tertiary education, you should skip directly to line 44. In writing the introduction, you should also try and be more precise. You do not “discover the temporal dynamics of skills and differences in skills that correspond to workers’ earnings (lines 75 -76)”. Instead, you study how skills changed over time, use your model to predict future changes, and then study how much of the variance in earning is explained by differences in acquired skills. Furthermore, is “differentiate” the appropriate word in Q1 (line 83)? Are the expressions “patterns of skills across academia” and  “pattern of topics” correct in Q2? To predict the future, you exploit the variation over time of DWA, not across academia. Wouldn’t it be better a simple “How do the DWAs taught in a curriculum evolve over time?” or something like this? Lastly, do you really predict (=out of sample prediction) graduates’ earnings in your paper?Numbering the sections would help readers follow the article’s narration.Some of the evidence reported in the Supporting Information should be moved into the main text, while some figures could be added in an Appendix. For instance, part of the discussion on how you built and selected the sample used to analyze how the DWA described earning variation should be reported in the main text (line 315). The details reported now are not sufficient to understand your empirical strategy.Kolmogorov test to select subsets of observations. You lost me here. As far as I know, the K-S test is a test of dominance between two distributions. How do you use it to select single observations?  You wrote in the Supporting Information: “It is possible that a subset of earnings records of a FOS does not effectively represent the distribution of the entire population. We perform the Kolmogorov-Smirnov (KS) statistical test for the subset population against the entire population. (lines 102-103” Specifically, how do you group observations into subsets? Which rule are you followings? I suggest using some tests to detect outliers in the earnings distribution rather than arbitrary group observations and compare their “performance” against the “population.”How exactly do you model average earnings? Your econometric models should be reported in the Supporting Information if you prefer. As it is, I do not understand how you add DWA to the earning equations: in lines 316-321, you say that you add DWAs across fields weighted (?) by propensity scores- but arent DWAs dummies at this point of the paper? – and then interacted with FOS. You lost me here. If I understood correctly, you are using dummies, specifically, a dummy for each DWA. This means that you are adding to a model with 2872 observations, 2070 DWA (SI, line 48) dummies, and 2070*47 (= n° FOS) and running a LASSO procedure to choose the best specification. Am I correct? If so, please try to be more precise in explaining this procedure, and provide some more details, possibly non-technical. For instance, is there any regularity in excluded DWA? Can some of these steps be interpreted economically?One of the ceteris paribus that you would like to add to your analysis of how much earning variation is explained by DWA detected from syllabi are local labor market conditions. Thus you should probably add to your models in Figure 4B also the state or geographical area of the university fixed effects.Which variables are included in the Baseline in figure 4C?In identifying DWAs that correspond to increased earnings after graduation (lines 362-374), you are implying that you have unbiased coefficients, which may not be the case in your setting. You must be aware that many potential threats (omitted variables, collinearity, etc.) may bias the coefficients, and warn the reader to take your results cautiously since they are conditional correlations.You have no limit to the number of figures, so why did you group them? For instance, Figure 4A should be named Figure 4 and inserted around line 293; Figure 4B should be called Figure 5  and reported around line 340, and so on. This will allow you to put titles to each panel of your figure, which actually are missing, and avoid using a general but ineffective title like that of Figure 4.

Minor points:

Line 198: what do you mean by “ most prevalent”?Line 354: you probably mean: “ by increasing the “share” of the earnings explained,” not the “variation.”I would use the term predict in predicting something out of the sample. When you study the association between sets of variables, you are explaining, describing, etc. Specifically, in the section predicting graduate earnings, you are looking at how much DWAs explain variation in earnings.Line 242: “Predicting educational trend” what do you mean by the word “educational”?Please, report all the coefficients, statistics, and information of the regressions behind SI Table S2; they may be informative for the reader.Congratulations on the work so far; I look forward to reading the revision.

We look forward to receiving your revised manuscript.

Kind regards,

Simona Lorena Comi

Academic Editor

PLOS ONE

Journal Requirements:

“This research is supported in part by the University of Pittsburgh Pitt Momentum Fund (MRF) and the Center for Research Computing (MRF). This work has been supported (in part) by Grant # 2109-33808 from the Russell Sage Foundation (MRF & SHB). Any opinions expressed are those of the principal investigator(s) alone and should not be construed as representing the opinions of the Foundation.”

4. Please update your submission to use the PLOS LaTeX template. The template and more information on our requirements for LaTeX submissions can be found at http://journals.plos.org/plosone/s/latex.

Reviewers' comments:

Reviewer's Responses to Questions

**Comments to the Author**

1. Is the manuscript technically sound, and do the data support the conclusions?

Reviewer #1: Yes

2. Has the statistical analysis been performed appropriately and rigorously? 

Reviewer #1: Yes

3. Have the authors made all data underlying the findings in their manuscript fully available?

Reviewer #1: Yes

4. Is the manuscript presented in an intelligible fashion and written in standard English?

Reviewer #1: Yes

5. Review Comments to the Author

Reviewer #1: PONE-D-22-21083

Connecting Higher Education to Workplace Activities and Earnings

Reviewer’s report

The evaluated paper addresses a relevant and current topic. The discussion section is well constructed and summarizes quite well the objectives and achievements of the article. I leave some comments hoping that they will be useful for the improvement of the manuscript.

1. The introduction should be expanded to better contextualize the paper. Please, cite works that have used the algorithm suggested in the paper (or text mining algorithms in general) in the context of the labor market and personnel selection.

2. The general and specific objectives, the research questions, and the novelty of the study should be better exposed in the first pages of the manuscript.

3. I wonder if the knowledge and skills currently described by the U.S. Department of Labor are connected to the actual requirements of the jobs (i.e., employers’ demands from job seekers). To identify potential curricular gaps, the ideal would also be to connect the course contents (knowledge and skills gained in college) with the demands of employers (for example, reviewing job offers posted on online job portals through text mining methods). In the discussion section, the authors should give their arguments in this regard.

4. K-means clustering is a classical way for text categorization, but the clustering algorithm should be better explained in the manuscript. In particular, how were the degrees and universities in figures 2 and 3 grouped?

5. On Lines 263 to 265, the authors state that: “We run several ordinary least squares (OLS) regressions to predict Δrdwa,f using various models of DWA propensity scores and inter-DWA relationships.” However, I do not get to see the results in the documents delivered. Typically, in the statistical literature, "propensity scores" refer to the probabilities predicted by a logistic regression model. Do they have the same interpretation in the context of the article? This needs clarification.

Minor issues

6. The wording of lines 322-340 should be improved and better explain the LASSO methodology.

7. Define “inconsistent student achievement.”

8. Explain better the intention of Figure 1.

6. PLOS authors have the option to publish the peer review history of their article (what does this mean?). If published, this will include your full peer review and any attached files.

Reviewer #1: No

---

## [Author Response · Author response to Decision Letter 0]

22 Dec 2022

RESPONSE TO REVIEWERS

We thank the reviewers for their critical and extensive assessment of our work "Connecting Higher Education to Workplace Activities and Earnings” submitted for consideration for publication PLOS ONE - PONE-D-22-21083. In the following we address their concerns point by point. We also specify revisions to the main paper and the Supplementary Materials. We were able to address all of the reviewers’ concerns and we feel like the manuscript is significantly improved.

REVIEWER COMMENTS

Editor (Remarks to the Author):

Introduction. I agree with reviewer 1 that the introduction of the paper is not very effective. You start very far from your empirical exercises- even mentioning America’s position in the global economy. Then, you pose questions to which you provide no answer: “ what educational foundations best enable students to achieve their career goals?” You look at entry wage, not careers. Another example: “How do academic majors change their curriculum over time?” while you describe some changes, do not explain how. And so on. Please, be more focused and effective in introducing your empirical study; after a succinct (3-4 lines) about the importance of tertiary education, you should skip directly to line 44. In writing the introduction, you should also try and be more precise. You do not “discover the temporal dynamics of skills and differences in skills that correspond to workers’ earnings (lines 75 -76)”. Instead, you study how skills changed over time, use your model to predict future changes, and then study how much of the variance in earning is explained by differences in acquired skills. Furthermore, is “differentiate” the appropriate word in Q1 (line 83)? Are the expressions “patterns of skills across academia” and “pattern of topics” correct in Q2? To predict the future, you exploit the variation over time of DWA, not across academia. Wouldn’t it be better a simple “How do the DWAs taught in a curriculum evolve over time?” or something like this? Lastly, do you really predict (=out of sample prediction) graduates’ earnings in your paper?

Authors: We thank the editor for the feedback and suggestions. We have made significant changes in the Introduction to be more focused on our study. Specifically, we have removed some background about U.S. education, jobs and earnings, and added introduction for methods and techniques (e.g., natural language processing, text mining and network principles) used in the study, including references. We have also revised research question 2 to the following:

Q2. How do the DWAs taught in a curriculum or field of study evolve over time?

Can the relationships between pairs of skills across all of academia help to predict the skill evolution?

For the out-of-sample prediction question, we did perform 5-fold cross validation for both sections. We describe this analysis in the caption of SI Figure S8. But, we agree that this information was not readily apparent in the main text and we have updated the fifth paragraph of Section 3.3 to clearly state that the models are evaluated with 5-fold cross validation. We also mention it in the Introduction.

Numbering the sections would help readers follow the article’s narration.

Authors: We thank the editor for the suggestion. We have added numbers to the section titles.

Some of the evidence reported in the Supporting Information should be moved into the main text, while some figures could be added in an Appendix. For instance, part of the discussion on how you built and selected the sample used to analyze how the DWA described earning variation should be reported in the main text (line 315). The details reported now are not sufficient to understand your empirical strategy.

Authors: We thank the editor for the comment. We provide more details in our response below, but our sample is simply the College Scorecard earnings records for FOS-university-year combinations with at least 10 syllabi in the OSP data. . We have updated the first paragraph of Section 3.4, SI Section 4 and added the following sentences to the main text to better explain our strategy:

“Though large, the OSP course syllabus data is not distributed evenly across fields-of-study and institutions. Some fields and institutions have much less course syllabi. Thus, to sufficiently estimate work activities taught in a FOS at a university, we limit earnings records for FOS (in an institute) that have at least 10 course syllabi; and perform Kolmogorov-Smirnov statistical test to make sure the remaining earnings records representative for the entire population of the field at the institute (more details on the selection process and criteria are in SI Section 4)”.

Kolmogorov test to select subsets of observations. You lost me here. As far as I know, the K-S test is a test of dominance between two distributions. How do you use it to select single observations? You wrote in the Supporting Information: “It is possible that a subset of earnings records of a FOS does not effectively represent the distribution of the entire population. We perform the Kolmogorov-Smirnov (KS) statistical test for the subset population against the entire population. (lines 102-103” Specifically, how do you group observations into subsets? Which rule are you followings? I suggest using some tests to detect outliers in the earnings distribution rather than arbitrary group observations and compare their “performance” against the “population.”

Authors: We want to clarify our intention of using KS test. We did not use the Kolmogorov test to select subsets of observations. To sufficiently estimate work activities taught in a FOS at a university, we apply a filtering process which limits earnings records for FOS (in an institution) that have at least 10 course syllabi; therefore, some earning records are removed from the FOS when taken in aggregate. We want to test whether the remaining records are representative for the entire population (i.e., all the earning records) for the FOS at the institute.

We thank the editor for the questions. We have changed “a subset of earnings records” to “the remaining earnings records” (after the filtering process) to reduce the misunderstanding. We have updated the manuscript and SI as stated in the previous response.

How exactly do you model average earnings? Your econometric models should be reported in the Supporting Information if you prefer. As it is, I do not understand how you add DWA to the earning equations: in lines 316-321, you say that you add DWAs across fields weighted (?) by propensity scores- but arent DWAs dummies at this point of the paper? – and then interacted with FOS. You lost me here. If I understood correctly, you are using dummies, specifically, a dummy for each DWA. This means that you are adding to a model with 2872 observations, 2070 DWA (SI, line 48) dummies, and 2070*47 (= n° FOS) and running a LASSO procedure to choose the best specification. Am I correct? If so, please try to be more precise in explaining this procedure, and provide some more details, possibly non-technical. For instance, is there any regularity in excluded DWA? Can some of these steps be interpreted economically?

Authors: We agree that we should clarify our modeling procedure. Our earnings data comes from the US Department of Education College Scorecard database which details the median earnings of graduating classes by FOS, institution, and year. College Scorecard earnings data is described in Section 2 - Materials and Methods “College Scorecard provides information about post-college earnings including median earnings of graduates working and not enrolled after completing highest credential in their first and second years for the two graduation cohorts of years 2016 and 2017.”

Another clarification is that the variables representing DWAs are not dummies. Instead, the variables are the propensity scores of DWAs, which vary by university, and even by FOS at a specific university. The propensity score of each DWA for a FOS (r_f(dwa)) and for a university (r_u(dwa)) are explained in Equation (1). These propensity scores are averages of document similarity scores based on language embeddings (i.e., DWA text compared to syllabus text) and are real-valued between 0 (dissimilar) and 1 (similar).

For the models involving DWAs, we perform LASSO feature selection for 2070 DWA features. LASSO is one of the most popular methods for feature selection; it minimizes the residual sum of squares subject to the sum of the absolute value of coefficients being less than a constant. This constraint tends to “regularize” large models by producing some 0 coefficients when variables are co-linear. In other words, the LASSO feature selection helps us narrow the set of DWAs down to the ones that improve the prediction of earnings. In the end, 331 DWA variables remain in the model.

Similarly, we perform LASSO feature selection for 53,820 (2070*26) FOS*DWA features, which reduces to 594 features. 26 is the number of the remaining FOS after the filtering process explained in SI lines 98-100 “Furthermore, we select FOS that have at least 30 earnings records across institutions for prediction tasks, resulting to the remaining 2601 earnings records in 26 FOS at 343 institutions (see Table S1 for details of numbers of observations of FOS in our analysis before and after filtering).”

This has now been clarified in the second paragraph of Section 3.4:

“LASSO [30] is one of the most popular methods for feature selection; it minimizes the residual sum of squares subject to the sum of the absolute value of coefficients being less than a constant. This constraint tends to “regularize” large models by producing some 0 coefficients when variables are co-linear. In other words, the penalty factor determines how many features are retained; using cross-validation to choose the penalty factor helps assure that the model will generalize well to future data samples.”

One of the ceteris paribus that you would like to add to your analysis of how much earning variation is explained by DWA detected from syllabi are local labor market conditions. Thus you should probably add to your models in Figure 4B also the state or geographical area of the university fixed effects.

Authors: We thank the editor for raising this critical question. We have come up with a solution to measure the variation across geographical areas. We group universities together based on their divisions (e.g., New England Division, West North Central Division), using U.S. Geographic Levels. These groups are represented using indicator variables in the regression analyses. We have added a discussion about this new variable at the end of the first paragraph of Section 3.4, and the new results have been reported in the second paragraph of Section 2.4 and SI Figure S11B.

Which variables are included in the Baseline in figure 4C?

Authors: The baseline models only use average earnings of the FOS (there are no covariance). Explained in Lines 350-352 “the DWA models perform significantly better than the baseline models in terms of RMSE (i.e., comparing the mean earnings of graduates of the target FOS; due to the limited data within FOS to perform cross validation, the school ranking is omitted)”; and the caption of Figure 4.C “...The baseline model is the mean earnings of graduates of that FOS…”.

In this revision, we have added geographical variables (GEO) to earnings prediction thanks to the editor’s suggestion. The baseline GEO model now includes geographical fixed effects. We have updated the first paragraph of section 3.5 “...Due to the limited earnings data within FOS to perform cross validation, the school ranking is omitted; the baseline models only include geographic variables (GEO)...”; and the caption of Figure 4.C “...The baseline GEO model only includes geographic variables…”

We have also updated section 3.5 and Figure 4C with the updated performances which involve GEO variables.

In identifying DWAs that correspond to increased earnings after graduation (lines 362-374), you are implying that you have unbiased coefficients, which may not be the case in your setting. You must be aware that many potential threats (omitted variables, collinearity, etc.) may bias the coefficients, and warn the reader to take your results cautiously since they are conditional correlations.

Authors: We thank the editor for raising the concerns. We have updated the third paragraph of the Discussion section to highlight the limitations of our work including selection bias, omitted variables, etc. We state:

Future work might build on our study by analyzing the causal implications of skill-level adjustments to course content. In particular, our study's approach is unable to address selection bias when students choose a university in which to enroll. But future work may study natural experiments that overcome this barrier. Potential examples include the hiring, firing, or retirement of new faculty, the creation of a new school or department, the emergence of a large employer (e.g., resulting from new tax credit), or large donations focused on specific learning outcomes.

You have no limit to the number of figures, so why did you group them? For instance, Figure 4A should be named Figure 4 and inserted around line; Figure 4B should be called Figure 5 and reported around line, and so on. This will allow you to put titles to each panel of your figure, which actually are missing, and avoid using a general but ineffective title like that of Figure 4.

Authors: We thank the editor for the suggestion. We have split Figure 4 into Figure 4, 5 & 6 and added the caption for each of the figures in the main text.

Line 198: what do you mean by “ most prevalent”?

Authors: We thank the editor for the question. It means “most common”. We have changed it to “most common”.

Line 354: you probably mean: “ by increasing the “share” of the earnings explained,” not the “variation.”.

Authors: We thank the editor for the suggestion. We have changed “variation” to “share”.

I would use the term predict in predicting something out of the sample. When you study the association between sets of variables, you are explaining, describing, etc. Specifically, in the section predicting graduate earnings, you are looking at how much DWAs explain variation in earnings.

Authors: We did perform out-of-sample prediction with 5-fold cross validation in our study. We compute RMSE and R-square for out-of-sample data points. We explain it in the first paragraph of Section 3.5 “We employ LASSO feature selection for DWAs and report model performance using 40 independent trials of 5-fold cross-validation to mitigate over-fitting.”.

We thank the editor for the question. We have revised the Introduction to emphasize the out-of-sample prediction evaluation with cross validation in our study.

Line 242: “Predicting educational trend” what do you mean by the word “educational”?

Authors: We thank the editor for the question. We have changed “Predicting educational trend” to “Predicting the change in taught skills”

Please, report all the coefficients, statistics, and information of the regressions behind SI Table S2; they may be informative for the reader.

Authors: We thank the editor for the suggestion. We have reported the coefficients and statistics for all the selected DWAs for the regressions in this GitHub folder. We have also added the sentence “The full list of all the selected DWAs including the coefficients and statistics can be found in this GitHub folder[13].” at the end of Section 3.5.

Reviewer #1 (Remarks to the Author):

The introduction should be expanded to better contextualize the paper. Please, cite works that have used the algorithm suggested in the paper (or text mining algorithms in general) in the context of the labor market and personnel selection. The general and specific objectives, the research questions, and the novelty of the study should be better exposed in the first pages of the manuscript.

Authors: We thank the reviewer for the suggestion. We have made significant changes in the Introduction to highlight the novelty and be more focused on our study. We have added introduction for methods and techniques (e.g., natural language processing, text mining and network principles) used in the study (including references) in the fourth paragraph of Introduction section.

I wonder if the knowledge and skills currently described by the U.S. Department of Labor are connected to the actual requirements of the jobs (i.e., employers’ demands from job seekers). To identify potential curricular gaps, the ideal would also be to connect the course contents (knowledge and skills gained in college) with the demands of employers (for example, reviewing job offers posted on online job portals through text mining methods). In the discussion section, the authors should give their arguments in this regard.

Authors: We thank the reviewer for the suggestion. We have added a discussion at the end of the fifth paragraph in Discussion section to address this:

“With the advances of text mining methods, one could extract skills described in course syllabi and job postings, and align those skills to connect educational contents with the demands of the labor market. There are some existing job skill taxonomies to describe job postings' requirements such as BG's or LinkedIn's proprietary skill taxonomies. Börner et al. (2018) analyze course syllabi and BG's job postings focusing on areas of Data Science and Data Engineering. They use BG's skill taxonomy instead of the one used by the U.S. Bureau of Labor Statistics to analyze skill discrepancies between research, education and jobs. Modeling job postings with NLP techniques has also been shown to be useful in understanding wage premia [33]. Although our study focuses on the work side of job seeking, we acknowledge that the demand from the employer side is also important to understand the holistic picture from skill offerings in higher education to skill demands in the labor market; which could benefit many applications such as identifying potential curricular gaps or recommending courses to meet jobs' requirements.”

K-means clustering is a classical way for text categorization, but the clustering algorithm should be better explained in the manuscript. In particular, how were the degrees and universities in figures 2 and 3 grouped?

Authors: We thank the reviewer for this clarifying question. The FOS are part of the syllabus metadata using the IPEDS FOS taxonomy used by the US Department of Education; that is, they are determined by experts and not through clustering techniques. K-means clustering requires us to predetermine a number of clusters that will be found (i.e., what is K?) and so we instead opt for a more flexible clustering approach. Hierarchical clustering produces a nested sequence of clusters like a tree (or dendrogram); which enables us to explore different clusters under different similarity thresholds. The groups of universities presented by different colors in Fig. 3 are defined by Carnegie classification; and these groups are mostly recovered by the clusters produced by the agglomerative hierarchical clustering method for most similarity thresholds. The novel OSP syllabus data enables us to present FOS and universities as embedding vectors using NLP techniques; these numerical vectors are used to construct the dendrograms with the clustering method.

We have added a brief explanation about the agglomerative hierarchical clustering in the first paragraph of section 3.2. “Hierarchical clustering [29] produces a nested sequence of clusters like a tree (also called a dendrogram). Agglomerative clustering builds the dendrogram from the bottom level, and merges the most similar (or nearest) pair of clusters at each level to go one level up. Hierarchical clustering can take any form of distance or similarity function, and the hierarchy of clusters enables us to explore clusters at any level of detail without the need of picking a number of topics as would be the case with K-means clustering.”

On Lines 263 to 265, the authors state that: “We run several ordinary least squares (OLS) regressions to predict Δrdwa,f using various models of DWA propensity scores and inter-DWA relationships.” However, I do not get to see the results in the documents delivered. Typically, in the statistical literature, "propensity scores" refer to the probabilities predicted by a logistic regression model. Do they have the same interpretation in the context of the article? This needs clarification.

Authors: We thank the reviewer for this clarifying question. In this report, we only apply one method to compute the DWA propensity scores for FOS and universities (explained in Section 3.1). We do have various models of inter-DWA relationships including Jaccard similarity, Cosine similarity, Euclidean distance, and Manhattan distance, and direct similarity from DWAs’ embedding vectors. The results are reported in SI Section 3A, and are mentioned in the main text in lines 269-272 “The results (see SI Section 3A) show that modeling DWA relationships based on their co-occurrence in syllabi with Jaccard similarity yields the best performances across all the models involving inter-DWA relationships. We discuss these results in the main text.”

To clarify, "propensity scores" refers to how similar the DWAs are to a syllabus using cosine distance between their language embedded vectors (explained in Section 3.1, Equation (1)), not the probabilities predicted by a logistic regression model. That is, you can consider propensity score to be the “inferred semantic distance” between DWAs and syllabi course descriptions.

To avoid the confusion, we have updated the fourth paragraph of Section 3.1. “While r_f (dwa) (the dwa propensity score of FOS f ) is the average over the similarity scores of that DWA across s ∈ S_f , r_u(dwa) (the dwa propensity score of university u) is the mean similarity score of that DWA across syllabi weighted by the estimated graduation rates (αf,u) of the syllabus’s field of study at that university.”; and the third paragraph of Section 3.3. “We run several ordinary least squares (OLS) regressions to predict ∆rdwa,f using the DWA propensity scores of FOS f (r_f (dwa)) and various models of inter-DWA relationships (described in Section 3.1).”

The wording of lines 322-340 should be improved and better explain the LASSO methodology.

Authors: We thank the reviewer for the suggestion. We have added these new sentences to the second paragraph of Section 3.4 to explain the LASSO methodology:

“LASSO [30] is one of the most popular methods for feature selection; it minimizes the residual sum of squares subject to the sum of the absolute value of coefficients being less than a constant. This constraint tends to “regularize” large models by producing some 0 coefficients when variables are co-linear. In other words, the penalty factor determines how many features are retained; using cross-validation to choose the penalty factor helps assure that the model will generalize well to future data samples.”

Define “inconsistent student achievement.” 

Authors: We have revised the Introduction section based on the previous comments. The revision does not include “inconsistent student achievement”.

Explain better the intention of Figure 1.

Authors: We thank the reviewer for the comment. The intention of Figure 1 to show evidence supporting the content of Section 3.1. Thanks to the novelty of OSP syllabi dataset, our work is the first attempt to connect workplace activities to higher education through course syllabi using The O*NET database that is produced by the US Bureau of Labor Statistics and details the labor market trends of workplace skills and activities by occupation. The visualizations in Figure 1 show that, using natural language processing technique (i.e., word embeddings), we are able to infer work activities in syllabi, revealing key differences among universities and fields of study. We have added the following paragraph as the beginning of the second paragraph in Section 3.1 to provide additional details about the process of modeling detailed work activities and course syllabi with word embeddings.

“We bridge this gap by detecting O*NET’s detailed work activities from syllabus course descriptions. Each syllabus in the OSP data contains a description of the course content, a list of references and recommended readings, and course requirements, such as assignments and exams. Given a syllabus, we extract the course’s title, description, and learning objectives from the text and concatenate them to form the course descriptions (details are in SI Section 1A). We apply word embeddings [24] and document similarity techniques from natural language processing to represent each DWA and syllabus as continuous vectors distributed in the same pre-trained language embedding space. Language embedding models enable us to describe the semantic similarity between two textual documents or sentences; here, we compare syllabus course descriptions to DWAs. We choose pre-trained fastText word embeddings from [25], which is constructed from all Wikipedia pages in 2017, the UMBC webbase corpus, and the statmt.org news data.”

---

## [Decision Letter · Decision Letter 1]

30 Jan 2023

PONE-D-22-21083R1

Connecting Higher Education to Workplace Activities and Earnings

PLOS ONE

Dear Dr. Frank,

Thank you for submitting your manuscript to PLOS ONE. After careful consideration, we feel that it has merit but does not fully meet PLOS ONE’s publication criteria as it currently stands. Therefore, we invite you to submit a revised version of the manuscript that addresses the points raised during the review process.

Overall, the reviewer and I are quite happy with the progress made so far. However, reviewer 1 still wants to see a few additional changes – minor points- to your article before recommending an unconditional acceptance. I agree with that assessment – you have responded very well to most of the issues raised, and as a result, the manuscript has taken a big step toward a final publication. However, I would still ask you to address the remaining comments of reviewer 1 in the last revision of your paper.

We look forward to receiving your revised manuscript.

Kind regards,

Simona Lorena Comi

Academic Editor

PLOS ONE

Journal Requirements:

Reviewers' comments:

Reviewer's Responses to Questions

**Comments to the Author**

1. If the authors have adequately addressed your comments raised in a previous round of review and you feel that this manuscript is now acceptable for publication, you may indicate that here to bypass the “Comments to the Author” section, enter your conflict of interest statement in the “Confidential to Editor” section, and submit your "Accept" recommendation.

Reviewer #1: All comments have been addressed

2. Is the manuscript technically sound, and do the data support the conclusions?

Reviewer #1: Yes

3. Has the statistical analysis been performed appropriately and rigorously? 

Reviewer #1: Yes

4. Have the authors made all data underlying the findings in their manuscript fully available?

Reviewer #1: Yes

5. Is the manuscript presented in an intelligible fashion and written in standard English?

Reviewer #1: Yes

6. Review Comments to the Author

Reviewer #1: Connecting Higher Education to Workplace Activities and Earnings

PONE-D-22-21083R1

The reviewer thanks the authors for the significant revision they have performed on the manuscript. In general, they adequately respond to all my concerns.

Some minor remarks:

- The sections and subsections are not numbered as the authors claim in their responses.

- The Materials and Methods Section (Lines 68 to 119) only shows materials (databases used). Actually, the methods appear starting from line 120 in the Results Section. Check this out.

- I keep seeing the term "propensity score" as confusing. It should be better clarified in the manuscript. Is it really the term that is used in computer science literature?

7. PLOS authors have the option to publish the peer review history of their article (what does this mean?). If published, this will include your full peer review and any attached files.

Reviewer #1: No

---

## [Author Response · Author response to Decision Letter 1]

3 Feb 2023

RESPONSE TO REVIEWERS

We thank the reviewers for their critical and extensive assessment of our work "Connecting Higher Education to Workplace Activities and Earnings” submitted for consideration for publication PLOS ONE - PONE-D-22-21083. In the following we address their concerns point by point. We also specify revisions to the main paper. We were able to address all of the reviewers’ concerns and we feel like the manuscript is significantly improved.

REVIEWER COMMENTS

Reviewer #1 (Remarks to the Author):

The sections and subsections are not numbered as the authors claim in their responses.

Authors: We have numbered the sections in the manuscript as suggested.

[Reviewer] The Materials and Methods Section (Lines 68 to 119) only shows materials (databases used). Actually, the methods appear starting from line 120 in the Results Section. Check this out.

Authors: We thank the reviewer for pointing it out. We have updated the section title as follows:

- “Materials and Methods” to “Materials”

- “Results” to “Methods and Results”

[Reviewer[ I keep seeing the term "propensity score" as confusing. It should be better clarified in the manuscript. Is it really the term that is used in computer science literature?

Authors: To reduce the confusion, we have changed “propensity score” to “relevance score” throughout the manuscript, and also added the definition of “relevance score” after the Equation (1) as follows “These relevance scores are a measure of how strongly the skill (i.e., dwa) is represented in a field or university”.

---

## [Decision Letter · Decision Letter 2]

14 Feb 2023

Connecting Higher Education to Workplace Activities and Earnings

PONE-D-22-21083R2

Dear Dr. Frank,

We’re pleased to inform you that your manuscript has been judged scientifically suitable for publication and will be formally accepted for publication once it meets all outstanding technical requirements.

Kind regards,

Simona Lorena Comi

Academic Editor

PLOS ONE

Additional Editor Comments (optional):

Reviewers' comments:

Reviewer's Responses to Questions

**Comments to the Author**

1. If the authors have adequately addressed your comments raised in a previous round of review and you feel that this manuscript is now acceptable for publication, you may indicate that here to bypass the “Comments to the Author” section, enter your conflict of interest statement in the “Confidential to Editor” section, and submit your "Accept" recommendation.

Reviewer #1: All comments have been addressed

2. Is the manuscript technically sound, and do the data support the conclusions?

Reviewer #1: Yes

3. Has the statistical analysis been performed appropriately and rigorously? 

Reviewer #1: Yes

4. Have the authors made all data underlying the findings in their manuscript fully available?

Reviewer #1: Yes

5. Is the manuscript presented in an intelligible fashion and written in standard English?

Reviewer #1: Yes

6. Review Comments to the Author

Reviewer #1: Thanks for the reviews. The manuscript has been substantially improved. I hope you can continue in this line of research.

7. PLOS authors have the option to publish the peer review history of their article (what does this mean?). If published, this will include your full peer review and any attached files.

Reviewer #1: No

---

## [Editor Report · Acceptance letter]

20 Feb 2023

PONE-D-22-21083R2 

Connecting Higher Education to Workplace Activities and Earnings 

Dear Dr. Frank:

I'm pleased to inform you that your manuscript has been deemed suitable for publication in PLOS ONE. Congratulations! Your manuscript is now with our production department. 

Kind regards, 

on behalf of

Professor Simona Lorena Comi 

Academic Editor

PLOS ONE